# Relaxation Phenomena in Low-Density and High-Density Polyethylene

**DOI:** 10.3390/polym16243510

**Published:** 2024-12-17

**Authors:** Viktor A. Lomovskoy, Svetlana A. Shatokhina

**Affiliations:** Frumkin Institute of Physical Chemistry and Electrochemistry, Russian Academy of Sciences (IPCE RAS), Leninskiy Prospekt 31, 119071 Moscow, Russia; lomovskoy49@gmail.com

**Keywords:** HDPE (high-density polyethylene), LDPE (low-density polyethylene), internal friction spectra, temperature–frequency dependencies, local dissipative processes, relaxation time, shear modulus defect, dissipation mechanisms

## Abstract

A study was conducted on the internal friction spectra and temperature dependencies of the frequency of free damped oscillatory processes excited in the investigated samples of low-density polyethylene (LDPE) and high-density polyethylene (HDPE) over a temperature range from −150 °C to +150 °C. It was found that the internal friction spectra exhibit several local dissipative processes of varying intensity, which manifest in different temperature intervals. The structure of the internal friction spectra and the peaks of dissipative losses are complex, as evidenced by the occurrence of sharp, locally temperature-dependent jumps in the intensity of dissipative losses observed throughout the entire temperature range. A theoretical analysis was performed to explore the relationship between the anomalous change in the frequency of the oscillatory process and the defect in the shear modulus, as well as the mechanisms of internal friction for the most intense dissipative loss processes identified in the internal friction spectra. A significant difference was revealed in the structure of the internal friction spectra of LDPE and HDPE in the temperature range of −50 °C to +50 °C. A comparison of the LDPE and HDPE samples was conducted based on changes in their strength characteristics, taking into account the locally temperature-dependent changes in the shear modulus caused by local dissipative losses observed in the internal friction spectra.

## 1. Introduction

Polyethylene is one of the most widely used polymers, found in various industrial applications. It is utilized in a diverse range of fields, from artificial implants for the human body to large-diameter pipes for transporting gas and oil, as well as films used for food and other packaging production. The requirements for the material characteristics necessary for the successful implementation and use of different types of polyethylene clearly vary significantly depending on the application area.

There are three main types of polyethylene: low-density polyethylene (LDPE), high-density polyethylene (HDPE), and linear low-density polyethylene (LLDPE). LDPE is characterized by a randomly arranged long-chain branched structure, with branches on branches, where the short-chain branches are uneven in length. Linear low-density polyethylene (LLDPE) features branches of uniform length that originate from the comonomer and are randomly distributed along the polymer chain. In contrast, HDPE virtually lacks both long and short branches. However, to achieve specific objectives, very small amounts of comonomer may be deliberately added to the HDPE product [1].

Certain features of the molecular structure are often a direct consequence of the polymerization process, specifically the type of polymerization and the initiator that facilitates the bonding of monomers into a polymer chain. Radical polymerization occurring under high pressure results in the formation of polyethylene chains that are highly branched (LDPE) with relatively long side chains. These polyethylene chains have poor crystallization properties, leading to a low density of the material. In contrast, polyethylene produced under low pressure through co-ordination polymerization, initiated by a catalyst containing a transition metal, is primarily linear with few or no branches. Such polymer chains crystallize easily, resulting in a high density of the material (HDPE).

There is a substantial amount of experimental research focused on studying the ultimate strength characteristics and physicomechanical properties of high-density polyethylene (HDPE) and low-density polyethylene (LDPE), as well as composite materials that incorporate these polymers [2,3,4,5,6,7,8,9,10,11,12,13,14,15,16,17]. The theoretical analysis of the obtained results is primarily based on the concepts of the “model of a homogeneous continuous medium”. However, alongside the investigation of macroscopic failure in polymer systems, an increasing number of studies are currently being conducted to address the fundamental problem of physicochemical mechanics, which examines the relationship in the following form: “chemical nature, composition, structure—physicochemical and physicomechanical characteristics—synthesis of materials with specified properties” [1,2,3,10,17,18,19,20].

The solution to this problem requires considering both the representations of the model of a homogeneous continuous medium and atomic-molecular models of the structure of the investigated systems. This is due to the fact that, as a result of external deforming influences on the investigated system, the reaction of this system to such influences will depend on the cumulative reaction of the individual structural–kinetic subsystems that collectively form the investigated system [21,22]. The reaction of each structural–kinetic subsystem manifests itself within its own temperature–frequency range, is quasi-independent of the reactions of other structural–kinetic subsystems, and is determined by its physical–mechanical and physical–chemical characteristics.

This is due to the fact that the elements of these subsystems have their own structure (link of a macromolecular chain, macromolecular segment, amorphous or crystalline phase, etc.) [10,23,24,25,26,27] and the transition of these elements from a thermodynamic and mechanical nonequilibrium state to an equilibrium state is determined by their transition function. These transition functions are one of the main factors leading to local dissipative processes in temperature and frequency, which are manifested in the internal friction spectra of the investigated systems as loss peaks of varying intensity located in different temperature ranges of this spectrum [10,24,25,28,29,30,31,32,33,34,35,36]. The combination of these transitional processes has a decisive impact on the performance of polymer material products under various operating conditions. The dynamic characteristics of these transitional processes are assessed based on the characteristics of the transition functions, which represent the system’s response to both internal changes within the system and external influences acting on it. The intensity of changes in the parameters of the transition functions can provide specific information about the structure of the investigated systems.

The presence of transitional processes from a nonequilibrium to an equilibrium state in a nonconservative system leads to the thermodynamic irreversibility of this process and, consequently, to the dissipation (internal friction) of part of the energy from external force influences within the studied system. A quantitative characteristic of energy dissipation in the system is the absorption coefficient, denoted as Ψ, which is related to other dissipative characteristics by the following relationship:(1)$$\Psi =\frac{\Delta W}{2\pi W}=Q^{-1}=tg\delta =\frac{\lambda }{\pi },$$
where ΔW and W represent the amounts of irreversibly dissipated energy and energy supplied to the system, respectively; $Q^{-1}$—the internal friction; δ—the phase shift angle between the external influence and the system’s response to that influence; and λ—the logarithmic decrement of the damping oscillatory process [37].

This allows for the acquisition of experimental internal friction spectra. Internal friction is defined as the process of irreversible energy dissipation within the volume of the investigated system due to external mechanical forces that deform the system [37,38]. The internal friction spectra consist of a set of local peaks representing dissipative losses manifested at various temperature ranges. In Equation (1), the logarithmic decrement of the oscillatory process is defined as follows:(2)$$\lambda =\frac{1}{n}\ln\frac{\varphi_{\max}}{\varphi \left(t\right)},$$
where n is the number of oscillations between the oscillation with amplitude $\varphi \left(t\right)$ and the oscillation with amplitude $\varphi_{\max}$ in the temporal representation of the oscillatory process.

For each temperature T of the investigation, there will be a corresponding temporal representation of the oscillatory process (as given in Equation (2)) and its own value of the logarithmic decrement λ, as well as its own value of the frequency ν of the oscillatory process excited in the investigated sample.

We previously published works [10,39], dedicated to the investigation of local dissipative processes observed in the internal friction spectra of HDPE, as well as the corresponding anomalous phenomena in the temperature dependencies of the frequency of free damped oscillatory processes associated with these dissipative processes. The study revealed four local dissipative processes of varying intensity, manifesting in different temperature ranges of these spectra (Figure 1).

The most complex and intense dissipative process is the $\beta_{k}$ process. This process is determined by the mobility of structural elements in the amorphous phase adjacent to the boundaries of various crystalline formations in the amorphous-crystalline structure of polyethylene (PE), as well as the oscillatory mobility of chain segments in the crystalline phase. It can be decomposed into several overlapping dissipative processes with respect to temperature, which is corroborated by data from studies conducted using other methods [40]. The second intense peak of dissipative losses, denoted as β, is located on spectrum $\lambda =f\left(T\right)$ in the temperature range from −140 °C to −90 °C. Like the $\beta_{k}$ process, it is also a complex process. The structural mechanism of this dissipative process is defined by the mobility of chain links in the amorphous phase and, similar to the $\beta_{k}$ process, is characterized by the relaxation mechanism of internal friction. The process associated with the glass transition of the amorphous phase of PE (α process) does not manifest distinctly on spectrum $\lambda =f\left(T\right)$ due to its complete absorption by the low-temperature branch $\beta_{k}$ of the process, which characterizes the differences in conformational positions of the passing chains in the amorphous intercrystalline layers of PE. This, in turn, leads to the formation of several structurally kinetic subsystems composed of similar structural units [2,10,15,19,26,41,42]. Additionally, on spectrum $\lambda =f\left(T\right)$ of the freely damped oscillatory process, another region with very weak intensity of local dissipative processes is identified as the μ process, observed in the temperature range ∼−70 °C (Figure 1).

The aforementioned observations pertain to HDPE, while LDPE has not been previously investigated by us. The previously presented internal friction spectra for LDPE, obtained using a horizontal torsional pendulum method with free damped oscillations, do not correspond to relaxation spectra with fine structure. The experimental data were collected over large temperature intervals (10–20 K), which resulted in the observation of broad maxima of dissipative losses [23,43], where such complex structures, as obtained in this study, were not detected. Furthermore, for the first time in this experiment, the temperature dependence of the frequency $\nu =f\left(T\right)$ of the free damped oscillatory process was obtained, and a theoretical analysis of the relationship between the anomalous change in the frequency $\nu =f\left(T\right)$ of the oscillatory process and the defect in the shear modulus $\Delta G=f\left(T\right)$ was conducted.

The relevance and practical significance of this work also lie in the fact that, as a result of the obtained experimental results and the theoretical analysis of these data, it is possible to develop recommendations for the use of products that include components made from the studied polymer systems in various climatic zones: from conditions of negative temperatures (operation of products in northern Arctic regions) to high temperatures (above the upper limit of the general industrial temperature range for operation).

This work presents an experimental study and theoretical analysis of all possible local dissipative processes observed over a wide temperature range from −150 °C (achieved by cooling with liquid nitrogen) to +150 °C, based on the spectra of $\lambda =f\left(T\right)$ and the temperature dependencies of the frequency of $\nu =f\left(T\right)$ free damped oscillatory processes for high-density and low-density polyethylene. The physical–mechanical (shear modulus defect) and physicochemical (activation energy of dissipative processes, discrete relaxation times, and determination of the mechanisms of each dissipative process—relaxation, phase, and hysteresis) characteristics of these local dissipative processes have been identified, along with the changes in these characteristics depending on structural modifications. This is reflected in the relaxation and structural microinhomogeneity of each of the identified local dissipative processes.

## 2. Experimental

### 2.1. Materials

The materials used in this study were polyethylene granules of the following industrial grades: LDPE 10803-020 (Kazanorgsintez, Kazan, Russia) and HDPE 277-73 (Kazanorgsintez, Kazan, Russia) (Table 1).

The samples were prepared as follows: the mold containing the polymer charge was placed in the press, with the mold temperature set at +24 °C. To prevent the polyethylene from sticking to the mold surface during pressing and to achieve a smoother surface, a polyimide film was used as a spacer between the sample and the mold. The press plates were brought together so that the polyethylene was subjected to a pressure of no more than 0.4 MPa, and the mold was heated for 15 min to a holding temperature of 160 °C. Subsequently, at the holding temperature, the sample was kept under a pressure of 4.9 MPa for an additional 5 min. After this, the samples were cooled to a temperature of 20 to 25 °C by wrapping the sides of the mold with a cold wet cloth. The cooling time to approximately +23 °C was about 20 min.

As a result, circular plates with a thickness of 1 mm were obtained. From these plates, rectangular samples with a thickness of 1 mm, a length of 65 mm, and a width of 5 mm were subsequently cut out.

The types of polyethylene differ in their degree of branching; for every 1000 carbon atoms, high-density polyethylene has 4 to 6 branches, while low-density polyethylene has more than 15 side branches (methyl, butyl, and ethyl groups). The varying content of methyl groups accounts for the differences in density and degree of crystallinity of the polymer [6,44]. The branching of macromolecules in low-density polyethylene (LDPE) leads to the formation of a loose, partially crystalline structure, which consequently results in a decrease in the density of the polymer [5].

### 2.2. Methods

Measurements using differential scanning calorimetry (DSC) were conducted with a DSC Q100 unit of Intertech Corporation (USA) at a rate of 5 °C/min at an argon current of 50 mL/min.

The experimental temperature dependencies of dissipative losses (internal friction spectra $\lambda =f\left(T\right)$) and the temperature dependencies of the frequency changes $\nu =f\left(T\right)$ of the free damped oscillatory process excited in the sample of the investigated material were obtained using a setup that consists of a horizontally executed pendulum structure (Figure 2) [45,46]. This research method is based on displacing the test sample from its mechanical equilibrium state by a relative strain $\varepsilon_{0}\approx 10^{-4}$ in an isothermal mode $T_{i}=const$ for each temperature in the range from −150 °C to 450 °C (123 to 723 K). The heating rate was 2 °C/min (ω=2 °C/min).

The sample (3) with a rectangular cross-section is positioned horizontally and is secured using collet clamps to the supporting tube and the horizontal rod (5) of the inertial system. At the center of gravity of the inertial system, there is a conical core (6) that rests on a support capable of longitudinal displacement as the length of the sample changes with temperature variations. Balancing the inertial system at the point of attachment of the core is achieved by moving a counterweight (12). To reduce oscillation losses due to friction between the conical core and the support, the core is made of superhard steel, while the support is made of agate used in watch pivot. The excitation of torsional oscillations is performed using electromagnets (17) located on the supporting plate (18).

A lifting–moving device is used to install a thermal cryochamber, which consists of two sections: a heating element section and a refrigerant section connected to a Dewar vessel. Temperature regulation (thermostabilization of the sample at a specific temperature or temperature change at a certain rate) in the range of 20 to 450 °C is performed by a temperature controller according to an automatically programmed schedule with a constant cooling and heating rate for the sample being tested. The temperature-sensitive element for temperature regulation is a platinum–rhodium thermocouple with a diameter of 0.1 mm and a length of about 3 m (with a resistance of 25 Ohms at room temperature), located near the sample (wrapped around a ceramic rod). When operating in the temperature range of −150 to +20 °C, the sample was cooled with liquid nitrogen to a minimum temperature value. The heating rate was maintained at a standard rate of 2 °C/min according to GOST.

In this case, the logarithmic decrement λ of a damped oscillatory process can be used as a measure of internal friction (Figure 3). This logarithmic decrement is included in the equation that describes the oscillatory process as follows:(3)$$\varphi (t)=\varphi_{\max}\exp^{-\beta \theta }=\varphi_{\max}\exp\;\left[-\left(\frac{\lambda }{\pi }\right)\;t\right],$$
where $\varphi_{\max}$ и φ(t) are the amplitude (maximum) and current value of the torsion angle of the free end of the studied sample, respectively; β is the damping coefficient; θ is the period of the oscillatory process; and t is time.

The tested sample 1 (Figure 3a) is rigidly fixed in an immovable clamp, which, together with the sample, is placed in the thermocryochamber of the device. The research can be conducted either in an isothermal mode ($T_{i}=const$) or with a constant rate of temperature change $T_{i}$.At time $t_{0}$ (Figure 3b), an external torque $M_{ext}=M_{tor}$ is applied to the free end of the tested sample (Figure 3a) in the form of a unit delta function $\Delta \left(t\right)$, defined by the following relation:(4)$$\left\{\begin{array}{l} \Delta \left(t\right)=0\;\mathrm{by}\;t\neq t_{0}\;M_{ext}=0\\ \Delta \left(t\right)=1\;\mathrm{by}\;t=t_{0}\;M_{ext}=1 \end{array}\right.,$$This leads to the twisting of sample 1 around its longitudinal axis *Z* (Figure 3a) at time $t=t_{0}$ by an angle $\varphi =\varphi_{\max}$ (Figure 3a,c).For $t>t_{0}$, the external torque is removed, i.e., $M_{ext}=0$ (Figure 3b), and the tested sample 1 (Figure 3a) begins to undergo a free damped torsional oscillation process around the longitudinal axis Z (Figure 3c). The presence of internal friction in the material of the tested sample leads to a decrease in the amplitude of the oscillatory process in all subsequent periods of this oscillation (Figure 3c). The envelope curve (dashed line in Figure 3c–e) is described by an equation, where λ is the logarithmic decrement of this process, characterizing the rate of damping, and, consequently, the internal friction, i.e., the dissipation (conversion) of part of the energy from the external impact (after the sample is brought into a nonequilibrium mechanical and thermodynamic state—presence of the angle $\varphi =\varphi_{\max}$ в at time $t=t_{0}$ into thermal energy irreversibly dissipated within the volume of the tested sample).Changes in temperature and the subsequent equivalent impact on the tested sample lead to changes in the rate of the oscillatory process and, consequently, in the logarithmic decrement λ, the period θ of oscillations, and the frequency ν of oscillations. This allows the construction of experimental curves for the internal friction spectra $\lambda =f\left(T\right)$ and the temperature dependence of the frequency $\nu =f\left(T\right)$.The torsional oscillations arising in the tested sample induce shear strain across the cross-section of the sample $\gamma \left(T\right)$ (Figure 3d), which is in phase δ with the twisting angle φ(t) throughout the entire time span of the oscillatory process (Figure 3c,d). Just like the logarithmic decrement λ, the period θ of oscillations, and the frequency ν of oscillations, the shear strain also depends on temperature.The occurrence of deformations $\gamma \left(T\right)$ across the sample’s cross-section is caused by shear stresses $\sigma_{ij}$ in the tested sample. A phase shift δ arises between the deformation $\gamma \left(T\right)$ and the stress $\sigma_{ij}\left(t\right)$ which, like the logarithmic decrement λ, the period θ of oscillations, and the frequency ν of oscillations, depends on temperature. This phase shift angle characterizes the degree of dissipation of part of the external energy within the volume of the tested system and defines the relationship between the frequency ν of oscillations and the complex modulus of elasticity of the sample material in the tested system. Specifically, it determines the connection between the loss modulus (logarithmic decrement λ)—the imaginary part of the complex shear modulus—and the shear modulus G—the real part of the complex shear modulus. This, in turn, allows for the calculation of the shear modulus defect for each local dissipative process detected in the spectrum $\lambda =f\left(T\right)$ and the temperature dependence of the frequency $\nu =f\left(T\right)$ of the oscillatory process over the entire temperature range of the study.

## 3. Results and Discussion

### 3.1. Investigation Using Differential Scanning Calorimetry (DSC)

Figure 4 presents the results of experimental studies of HDPE and LDPE in determining the degree of crystallinity and melting temperature of the investigated systems, obtained using the DSC method.

The observed peak on the thermograms corresponds to the endothermic melting process of the crystalline phase of PE and is located in the range of 105 to 135 °C. The degree of crystallinity of sample χ was determined using relation 5 [10,48]:(5)$$\chi =\left(\frac{\Delta H_{m}}{\Delta H_{m}^{0}}\right)\cdot 100\%,$$

где $\Delta H_{m}$ is enthalpy of melting of PE samples; $\Delta H_{m}^{0}$ is thetheoretical value of polymer melting enthalpy with 100% degree of crystallinity $\Delta H_{m}^{0}=293\;\mathrm{kJ}/\mathrm{kg}$ [2,49]. The calculated values of crystallinity for the samples are presented in Table 1.

### 3.2. Investigation Using Relaxation Spectroscopy

In Figure 5, the internal friction spectra $\lambda =f\left(T\right)$ are presented in panel (a), and the temperature dependence of the frequency $\nu =f\left(T\right)$ of free damped torsional oscillations excited in the studied samples is shown in panel (b) for HDPE and LDPE within the temperature range from −150 °C to +150 °C.

In the spectra $\lambda =f\left(T\right)$, in the temperature range from −150 °C to +150 °C, PE exhibits two temperature regions of the most intense local dissipative processes (β and $\beta_{k}$). The most complex dissipative process is the process $\beta_{k}$. It is determined by the mobility of structural elements of the amorphous phase adjacent to the boundaries of various crystalline formations in the amorphous-crystalline structure of PET, as well as the vibrational mobility of segments of the chains in the crystalline phase. This loss peak can be decomposed into several overlapping dissipative processes with respect to temperature, which is confirmed by data from studies obtained using other methods [40] (Figure 5).

The second intense peak of dissipative losses—β, located on the internal friction spectrum $\lambda =f\left(T\right)$ within the temperature range from −140 °C to −90 °C, is also a complex process, similar to the $\beta_{k}$ process. The complexity of this process is manifested in the emergence of sharp, localized temperature spikes in the intensity of dissipative losses, observed on both the low-temperature and high-temperature branches of the β process. According to G.M. Bartenyev, the structural mechanism of this dissipative process is determined by the mobility of the $CH_{2}$ structural–kinetic elements [23].

In the temperature range from −50 °C to +50 °C, distinct loss peaks are observed in LDPE, while, in HDPE, these peaks are presumably absorbed by the low-temperature branch of the $\beta_{k}$ process. These loss peaks are designated in the spectrum as α-, $\alpha_{1}$-, $\alpha_{2}$-, and $\alpha_{3}$-processes, which are associated with the segmental mobility of the structural elements of macromolecules in the amorphous subsystem of the polymer. The internal friction spectra of LDPE enable the observation of these peaks, allowing for their separation (in the first approximation directly on the internal friction spectra) and the determination of their physicomechanical characteristics based on experimental data (internal friction spectrum, Figure 5, Table 2).

In Table 2 and Table 3 (provided below), the experimental and fundamental physical–mechanical and physicochemical characteristics, calculated based on the obtained experimental data, are presented for all processes of dissipative losses in LDPE and HDPE.

The calculation of physical–mechanical and physicochemical characteristics for the β and $\beta_{k}$ processes of dissipative losses was conducted based on the model representations of a standard linear solid. The solution to the differential equation of the standard linear solid in dynamic mode, taking into account the temperature-frequency dependence of the logarithmic decrement of the damped oscillatory process, is expressed as follows [10,21]:(6)$$\lambda_{i}=2\lambda_{i_{\;\max}}\frac{\omega \tau }{1+{\left(\omega \tau \right)}^{2}},$$
where $\lambda_{i}$ and $\lambda_{i_{\;\max}}$ are the current and maximum values of the logarithmic coefficient of the damped oscillatory process for the i-th dissipative process; $\tau \equiv \tau_{i}=\frac{\eta }{G_{1}}$ is relaxation time of the i-th subsystem, causing the appearance of the dissipative loss peak on the spectrum $\lambda =f\left(T\right)$.

According to Deborah’s frequency–time relationship, $\lambda_{i}$ reaches its maximum at the peak of losses (when $\left(\lambda_{i}=\lambda_{\beta_{k\;\max}}\right)$) under the condition specified by Equation (6).
(7)$$\omega \tau_{i}=1,$$
where the relaxation time $\tau_{i}$ corresponds to the relaxation time at the peak of dissipative losses in the spectrum $\lambda =f\left(T\right)$ and is defined by the Arrhenius equation:(8)τ=τ0expERT,
where E is the activation energy of the dissipative process; $\tau_{0}\approx 1.6\cdot 10^{-13},\;s$ is the theoretical value of the pre-exponential factor characterizing the oscillatory process of a relaxing particle at the bottom of the potential well for polyethylene [23], and R is the gas constant.

The frequency of the oscillatory process ν (determined experimentally from the dependence $\nu =f\left(T\right)$) is related to the angular frequency ω by the relationship ω=2πν. This allows for the determination of the relaxation time $\tau =\tau_{\max}$ at the peak of local dissipative losses $\lambda_{\max}$ based on the corresponding frequency value $\nu =\nu_{\max}$ and on the temperature dependence $\nu =f\left(T\right)$. For example, for the β-dissipative process observed in the internal friction spectrum for both polyethylene HDPE and LDPE:(9)$$\begin{array}{l} \tau_{\beta }=\frac{1}{2\cdot \pi \cdot 3.94}=0.040\;s\;\text{-}\;\mathrm{for}\;\mathrm{LDPE};\\ \tau_{\beta }=\frac{1}{2\cdot \pi \cdot 3.47}=0.046\;s\;\text{-}\;\mathrm{for}\;\mathrm{HDPE}. \end{array},$$

The activation energy of this process is determined from the Arrhenius dependence of the relaxation time τ on temperature (Equation (8)), taking into account Equation (9):(10)Uβmax=RTβmaxlnτβmaxτ0==8.314·157*·ln0.0401.6·10−13=34257.19 Jmol≈34.3 kJmol- for LDPE;Uβmax=8.314·165*·ln0.0461.6·10−13≈36.3 kJmol- for HDPE.*The temperature is given in K.

The temperature dependence of the $\nu =f\left(T\right)$ of the free damped oscillatory process excited in the studied system allows for the determination of the temperature change of the shear modulus $G\left(T\right)$ of the material from which the sample is made, across the entire temperature range of investigation. The experimental dependencies $\nu =f\left(T\right)$ (Figure 5b) show that, in certain temperature intervals, where local dissipative processes are observed as peaks in the loss spectra $\lambda =f\left(T\right)$, there is an anomalous change in the frequency of free damped oscillations in the dependencies $\nu =f\left(T\right)$ and, consequently, in the shear modulus G. In this case, a significant deviation of the experimental curve from the proportional theoretical temperature dependence $G=f\left(T\right)$ or $\nu =f\left(T\right)$ is observed. To describe this anomaly, the concept of the shear modulus defect ΔG or frequency defect Δν is introduced.

To determine the mechanism of internal friction associated with the dissipative processes identified on the internal friction spectrum $\lambda =f\left(T\right)$, a calculation of the magnitude and sign of the shear modulus defect was performed based on the temperature dependence of the frequency of the oscillatory process (see Figure 6) using the following relationship [21]:(11)$$\Delta G\left(T\right)=\frac{G_{0}\left(T_{0}\right)-G_{i}\left(T_{i}\right)}{G_{0}\left(T_{0}\right)}=\frac{\nu_{0}^{2}\left(T_{0}\right)-\nu_{i}^{2}\left(T_{i}\right)}{\nu_{0}^{2}\left(T_{0}\right)}.$$

Based on the experimental data and Equation (11), we obtain:$$\begin{array}{l} \Delta G_{\beta }=\frac{4.63^{2}-3.41^{2}}{4.63^{2}}=0.460\;\text{-}\;\mathrm{for}\;\mathrm{LDPE};\\ \;\Delta G_{\beta }=\frac{4.06^{2}-2.99^{2}}{4.06^{2}}=0.455\;\text{-}\;\mathrm{for}\;\mathrm{HDPE}. \end{array}$$

The shear modulus defect can have a positive value for dissipative processes of a relaxational nature and a negative value for dissipative processes of a non-relaxational nature. The obtained values of the shear modulus defect allow for a quantitative determination of the actual change in the strength characteristics of the studied materials, taking into account local temperature variations in the shear modulus caused by local dissipative losses introduced by each local dissipative process. These losses are manifested in the internal friction spectrum $\lambda =f\left(T\right)$.

We observe that, since the values of the shear modulus defect for LDPE are greater than those for HDPE, LDPE exhibits a lower ability to resist elastic deformation under external influences across the entire temperature range. The most significant changes are noted in the temperature range associated with the manifestation of $\alpha +\beta_{k}$ dissipative processes. This is related to the emergence of local inelastic phenomena within the polymer system. A positive value of the shear modulus defect indicates a relaxational nature of internal friction [10].

#### 3.2.1. Segmental Mobility of Polyethylene (α-Relaxation Process)

The main differences in the structural intricacies of the macromolecular chains of LDPE and HDPE are clearly manifested in the internal friction spectra of these systems (Figure 5 and Figure 7). For HDPE, the spectrum prominently exhibits two dissipative processes (β and $\beta_{k}$ processes), whereas, for LDPE, three dissipative processes (β, α, and $\beta_{k}$ processes) are present in the spectrum. Notably, the α dissipative process, associated with the segmental mobility of the macromolecular chains in the amorphous phase, is complex and consists of a combination of several processes (Figure 7).

The first point to note is that we observe two peaks, one of which (the α peak) absorbs the subsequent peak (the $\alpha_{1}$ peak). However, the separation of the peaks is quite noticeable in the internal friction spectrum, allowing for calculations to be performed for each of them. The intensity of these peaks decreases slightly with increasing temperature. At the same time, if we refer to Figure 8, where the spectrum has been decomposed into several local sub-peaks of dissipative losses through mathematical processing (using Gaussian distribution), we find that the largest contribution to the intensity comes from the very first α peak. The subsequent $\alpha_{1}$ peak is difficult to isolate using this method due to the extremely small difference in its intensity value. This means that the $\alpha_{1}$-peak contributes the least to the damping of the oscillatory process. The following peaks, $\alpha_{2}$ and $\alpha_{3}$, indicated in Figure 8 by green and blue lines, respectively, provide significant contributions, although they are much smaller in comparison to the α peak.

Thus, we conclude that the lowest-temperature structural subsystem contributing to the formation of the $\beta_{k}$-peak of dissipative losses in the spectrum responds at a temperature of −20 °C, with an activation energy of 49.2 kJ/mol (which corresponds to the activation energy value for segmental mobility obtained in earlier studies [15,23,42,50]), a relaxation time of 0.075 s, and represents the more mobile segments across all regions of the amorphous component. As we transition to higher temperatures, subsystems that experience greater mobility constraints begin to respond, leading to an increase in activation energy and relaxation time, as illustrated in Figure 9.

The increase in activation energy values and relaxation times for these processes confirms the previously made conclusions and indicates that, as the temperature rises, segments that were previously hindered in mobility become responsive. This may be related to their positioning within the structure of the polymer’s main amorphous phase: inter-lamellar and intra-lamellar amorphous phases, interphase layer, as well as the topology of the polymer structure.

#### 3.2.2. High-Temperature Transition ($\beta_{k}$-Relaxation Process)

In the internal friction spectrum (Figure 5a and Figure 7) of LDPE, the low-temperature branch of the $\beta_{k}$-relaxation process is completely absorbed by the high-temperature branch of the α-relaxation process. However, it is still possible to identify a peak and calculate the physicomechanical characteristics due to the clear manifestation of the maximum of this process. The spectrum of HDPE differs significantly from that of LDPE in that the $\beta_{k}$ process is more pronounced, absorbing the preceding α process to such an extent that it becomes extremely difficult to separate them.

The $\beta_{k}$ process for HDPE is shifted to the right in terms of temperature, into the region of higher temperatures (the difference is 5 °C). The intensity and frequency values are also greater for HDPE compared to LDPE (Table 2). However, despite the differences in the listed parameters, the activation energy values are nearly equal (70.6 kJ/mol for LDPE and 70.9 kJ/mol for HDPE), while the relaxation time is significantly lower for PEVP (Figure 10, Table 2).

Based on the obtained values, it can be hypothesized that a specific subsystem, such as a segment of the chain of a certain size, is responsible for the relaxation process in both materials. Furthermore, the difference in relaxation times at the same activation energy may be attributed to topological factors. This suggests that variations in the structure and configuration of the polymers influence the dynamics of relaxation, despite the similarities in their energetic characteristics.

#### 3.2.3. The Width of the $\alpha +\beta_{k}$ Peaks of Dissipative Losses

To evaluate the width of the $\alpha +\beta_{k}$ peaks of dissipative losses, the values of temperatures and relaxation times at these temperatures were determined according to relation (8) at λ=12λmax.

Based on the spectra (Figure 11) and tabular data (Table 4), it can be observed that the width of the $\alpha +\beta_{k}$ peaks at half-height changes insignificantly, while the relaxation microinhomogeneity of the Δτ processes differs significantly. For HDPE, this parameter is considerably higher, indicating a greater diversity of structural–kinetic subsystems responsible for the manifestation of dissipative processes within this temperature range.

The question of relaxation microinhomogeneity of processes and its relationship with the polymer structure, the nature, and mobility of subsystems is complex and constitutes a separate area of research.

## 4. Conclusions

It has been established that the internal friction spectra may exhibit either two or three intense local dissipative processes, depending on the density of the polyethylene system under investigation (HDPE and LDPE), as well as several weakly intense dissipative processes located in different temperature ranges of this spectrum. It has been demonstrated that these loss peaks, in turn, represent a combination of superimposed dissipative processes, which is manifested in the splitting of these loss peaks into their components.It has been demonstrated that the temperature dependence of the frequency of the oscillatory process induced in the investigated sample exhibits anomalous frequency changes, allowing for the determination of the magnitude and sign of the modulus defect and the establishment of the mechanism of internal friction for each dissipative process. It has been found that LDPE has a lower capacity for elastic resistance to external influences across the entire temperature range. The most significant changes are observed in the temperature region corresponding to the manifestation of $\alpha +\beta_{k}$ dissipative processes.In the internal friction spectra of LDPE in the temperature range from −50 °C to +50 °C, distinct loss peaks are clearly observed, whereas, for HDPE, these peaks are absorbed by the low-temperature branch of the $\beta_{k}$ process. As the temperature increases, there is an observed rise in the values of activation energy and relaxation times of these processes, which is associated with the arrangement of the responding subsystems and their topology. The relaxation microinhomogeneity in this temperature range is higher for HDPE, indicating a greater diversity of structural–kinetic subsystems responsible for the manifestation of dissipative processes within this temperature range.For the first time, all physical and mechanical characteristics (the defect modulus characterizing the region of local inelasticity under each peak of dissipative losses; the intensity of dissipative losses; the width of the temperature interval; and the temperature of the maximum value of the loss peak) as well as the physicochemical characteristics (activation energy, discrete relaxation time, and degree of relaxation microinhomogeneity) have been calculated based on experimentally obtained dependencies of internal friction spectra and temperature–frequency dependencies.It has been established that the nature of the occurrence of each of the high-intensity local dissipative processes observed in the internal friction spectrum is relaxation-based.

## Figures and Tables

**Figure 1 polymers-16-03510-f001:**
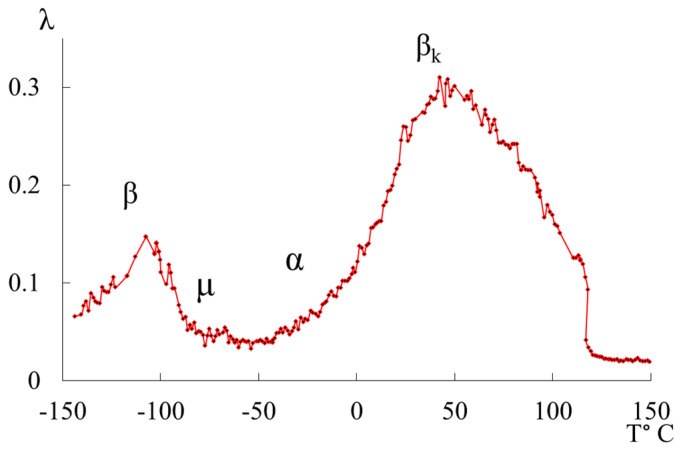
Internal friction spectrum $\lambda =f\left(T\right)$ of HDPE.

**Figure 2 polymers-16-03510-f002:**
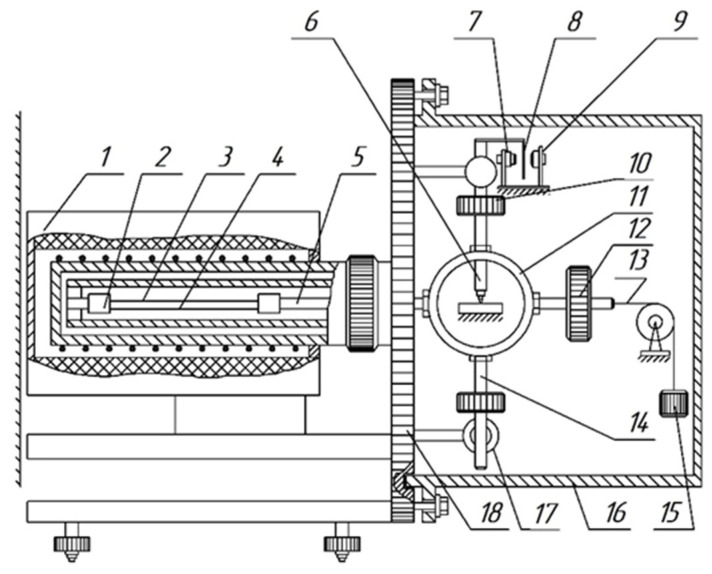
Schematic representation of a horizontal torsional pendulum. 1—Furnace housing; 2—collet; 3—sample; 4—substrates used in the high-temperature range; 5—horizontal rod; 6—core; 7–9—photoelectric transducer; 10—inertial weights; 11—shell; 12—counterweight; 13—tension string; 14—pendulum beam; 15—weight for tension; 16—vacuum cover; 17—electromagnets; 18—base plate.

**Figure 3 polymers-16-03510-f003:**
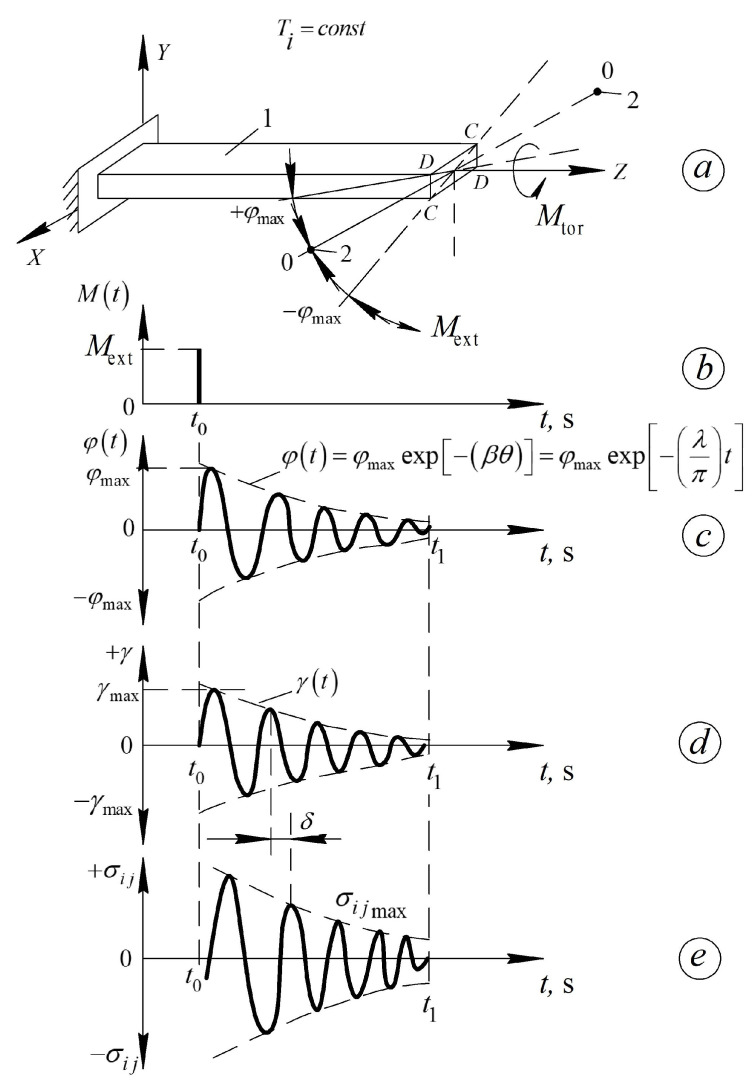
Diagrams outlining the freely damped oscillatory process induced in the studied material; (**a**) in isothermal mode T=const; (**b**) by pulse action. Sweep of the time dependence of the twist angle $\varphi \left(t\right)$ relative to the longitudinal axis Z of the specimen—(**c**). The deformation of the sample—$\gamma \left(t\right)$—(**d**) and the corresponding shear stresses $\sigma_{ij}$ occurring in the sample—(**e**). β—damping coefficient of the oscillatory process; θ—period of the vibration process. All other designations are defined below in the text of the article [21,47].

**Figure 4 polymers-16-03510-f004:**
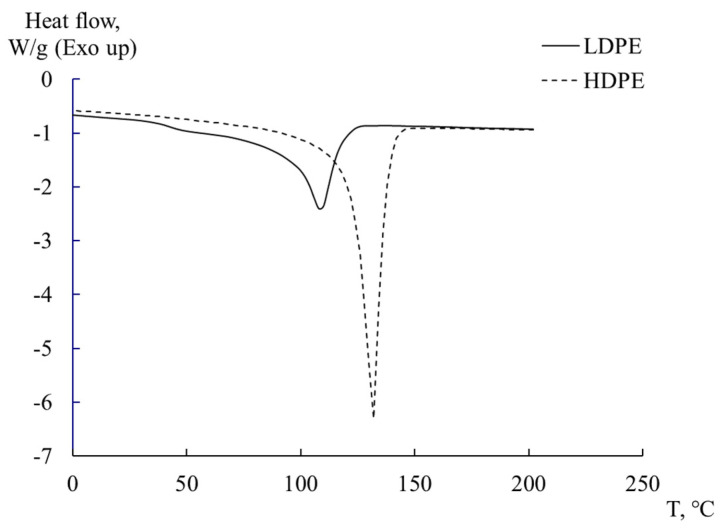
DSC thermograms for LDPE and HDPE.

**Figure 5 polymers-16-03510-f005:**
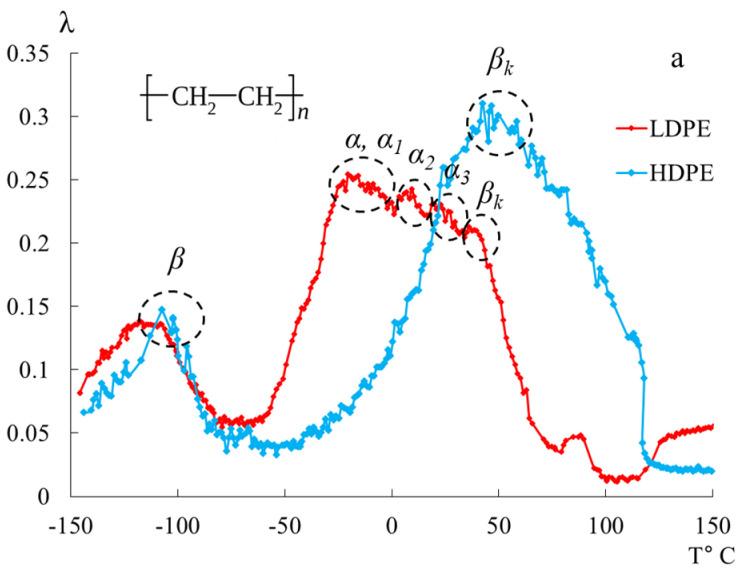
Internal friction spectrum $\lambda =f\left(T\right)$ (**a**); temperature dependence of frequency $\nu =f\left(T\right)$ (**b**) for LDPE (red line) and HDPE (blue line).

**Figure 6 polymers-16-03510-f006:**
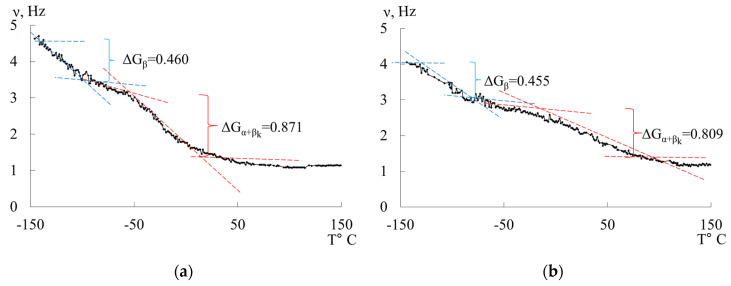
Temperature–frequency dependencies and a graphical example of determining the modulus defect ΔG for LDPE (**a**) and HDPE (**b**).

**Figure 7 polymers-16-03510-f007:**
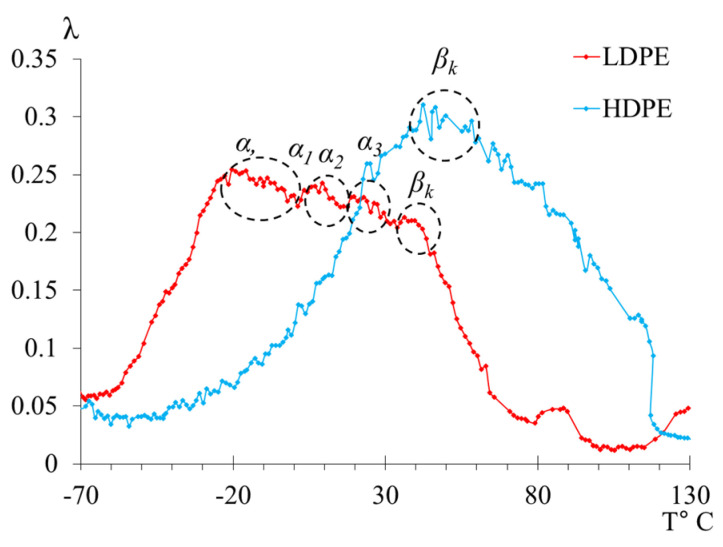
The internal friction spectrum of LDPE and HDPE in the temperature range from −70 °C to +130 °C.

**Figure 8 polymers-16-03510-f008:**
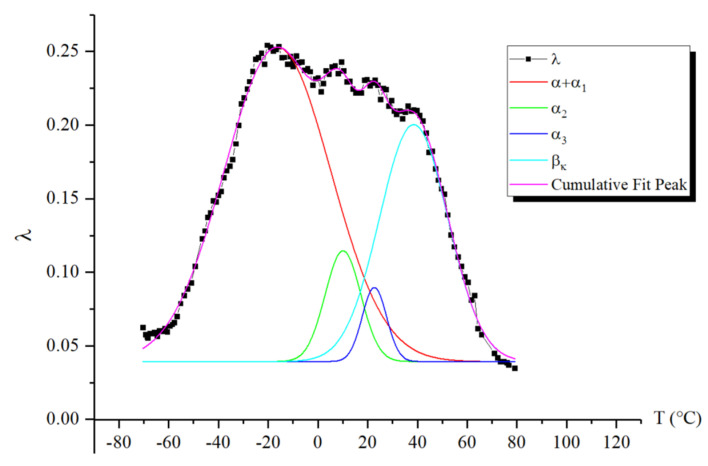
Decomposition of the α peaks and the $\beta_{k}$ peak of dissipative losses using a mathematical method based on Gaussian distribution.

**Figure 9 polymers-16-03510-f009:**
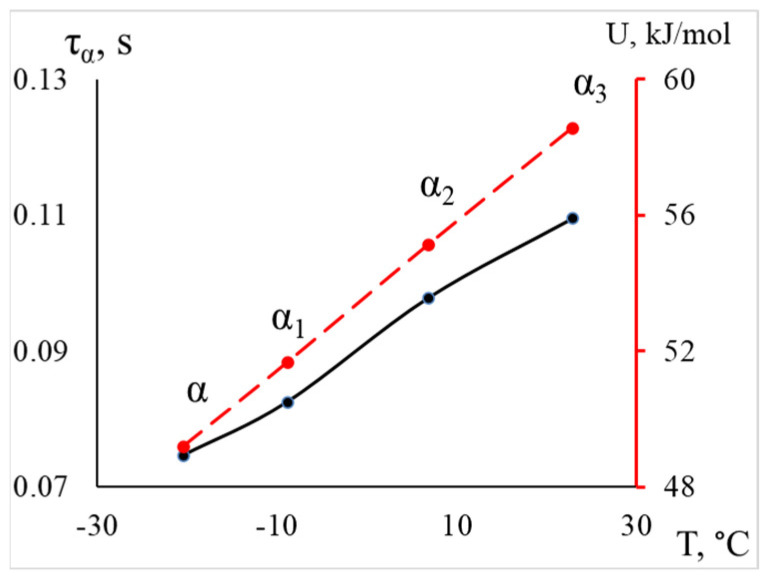
The dependence of the relaxation time and activation energy of α processes on temperature for LDPE is shown, with the activation energy values indicated by a red dashed line on the additional axis on the right.

**Figure 10 polymers-16-03510-f010:**
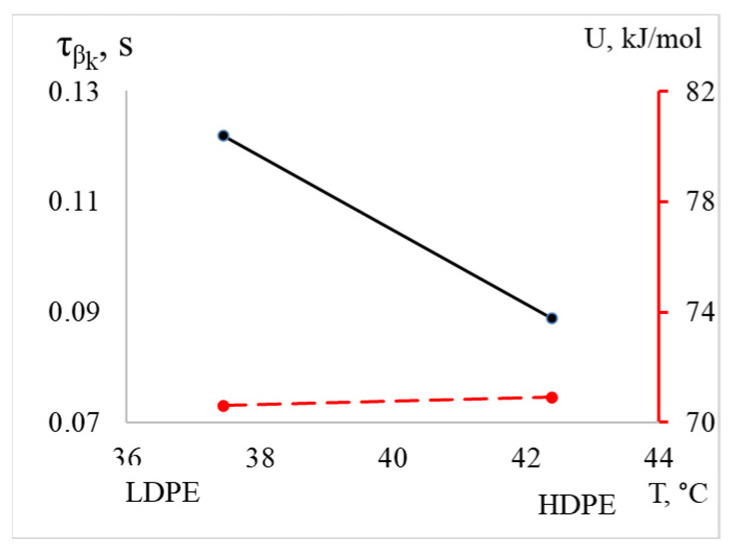
The dependence of relaxation time and activation energy of the $\beta_{k}$ processes on temperature for LDPE and HDPE is illustrated, with the activation energy values indicated by the red dashed line on the additional axis on the right.

**Figure 11 polymers-16-03510-f011:**
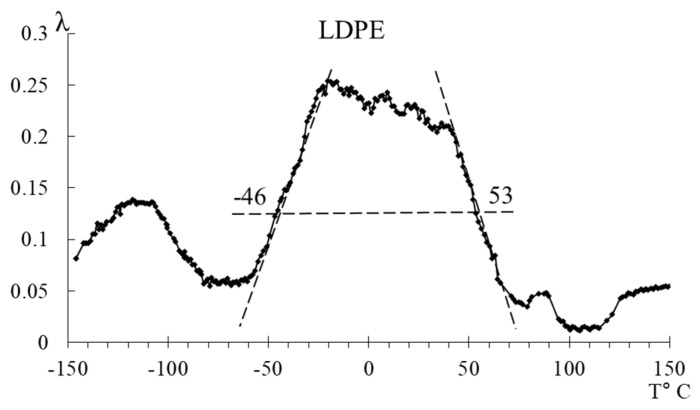
Schematic representation of the separation of $\alpha +\beta_{k}$ peaks of LDPE and HDPE.

**Table 1 polymers-16-03510-t001:** The main physicochemical and physicomechanical properties of LDPE. and HDPE.

PE Grade	LDPE 10803-020	HDPE 277-73
Density at 23 °C, g/cm^3^	0.920	0.957
Melt Flow Rate, g/10 min 190 °C/5.0 kg 190 °C/2.16 kg		
	-	17.0–25.0
	2	-
DSC Melting Point, °C	105	131
∆H_m_, J/g	73.5	157.1
Degree of crystallinity, χ, %	25	54

**Table 2 polymers-16-03510-t002:** The main physicomechanical and physicochemical characteristics for all processes of dissipative losses.

PE Grade	T_max_ (K)	T_max_ (°C)	$\lambda_{\max}$	$\begin{array}{l} \nu_{\max},\\ \mathrm{Hz} \end{array}$	*U*, kJ/mol	$\tau_{\max},s$	Nature of Process
β-process							Chain unit Relaxation mechanism of internal friction
LDPE	157	−116	0.134	3.94	34.3	0.040	
HDPE	165	−107	0.148	3.47	36.3	0.046	
α-process							Chain segment Relaxation mechanism of internal friction
LDPE	253	−20	0.254	2.13	49.2	0.075	
HDPE	-	-	-	-	-	-	
$\alpha_{1}$-process							
LDPE	264	−9	0.247	1.93	51.7	0.083	
HDPE	-	-	-	-	-	-	
$\alpha_{2}$-process							
LDPE	280	7	0.240	1.63	55.1	0.098	
HDPE	-	-	-	-	-	-	
$\alpha_{3}$-process							
LDPE	296	23	0.231	1.45	58.6	0.110	
HDPE	-	-	-	-	-	-	
$\beta_{k}$-process							Passing chains + oscillations of chain sections in the crystalline phase Relaxation mechanism of internal friction
LDPE	310	37	0.210	1.31	70.6	0.122	
HDPE	315	42	0.310	1.79	70.9	0.089	

**Table 3 polymers-16-03510-t003:** Experimental values of temperatures and frequencies, as well as calculated values of shear modulus defects for LDPE and HDPE.

PE Grade	T_max_ (°C)		The Range of Frequency Variation, Hz.		Shear Modulus Defect ΔG
	β-process				
LDPE	−146	−93	4.63	3.41	0.460
HDPE	−144	−85	4.06	2.99	0.455
	$\alpha +\beta_{k}$-process				
LDPE	−60	22	3.15	1.13	0.871
HDPE	−26	97	2.69	1.18	0.809

**Table 4 polymers-16-03510-t004:** Comparative experimental data on temperatures and calculated values of relaxation times, along with the corresponding ranges for $\alpha +\beta_{k}$ processes.

PE Grade	T_max_ (°C)		τ, s		Δτ, s	ΔT, °C
LDPE	−46	53	13.344	0.033	13.331	99
HDPE	7	103	84.334	0.001	84.333	96

## Data Availability

Data are available upon request to the corresponding author.

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
