# Peer review of "Relaxation Phenomena in Low-Density and High-Density Polyethylene"

_polymers, 2024, doi:10.3390/polym16243510_

Round 1

Reviewer 1 Report

Comments and Suggestions for Authors

this paper examines the relaxation phenomena in HDPE and LDPE by a free torsional pendulum. the results obtained highlight the existence of different relaxation phenomena in LDPE, in contrast to HDPE, showing a more regular behavior characterized by two main relaxation ranges. the paper is well written and presents interesting results. I only have to advise for some corrections:

1)      it is not completely clear how the difference observed in the high temperature relaxation between LDPE and HDPE correlates with the different molecular and/or crystalline structure

2)      in the framework of the presented discussion, DSC analysis does not provide any relevant information, and, in my opinion, ca n be removed.

Some minor issues:

1) row 200. as far as i understood, the authors performed isothermal scans? what has heating rate to do with it?

2) row 272 please edit the sentence which contains a repetition

3) row 347 it should not be "the current temperature change"

4) row 365 it is not completely clear how v correlates with G

5) table3 : the unit measures of deltaG are missing

6) row 454 please amend the acronyms PETN and PEVP

7) row 474: this should be table 4

Author Response

We thank the reviewer for their time and helpful comments - the feedback has greatly improved this article. Our responses to all comments are on file (reviewer comments are in bold). All changes to the text of the article are color-coded: green is what has been added; yellow is what is to be deleted.

Reviewer 2 Report

Comments and Suggestions for Authors

Review Report _ polymers-3326730-peer-review-v1

I have gone through the article entitled “Relaxation Phenomena in Low-Density and High-Density Polyethylene”. The authors have explained the different relaxation phenomena very nicely through in-depth analysis. However, these relaxation phenomena have also been discussed earlier by other authors for polymer system. Hence, the authors need mentioned that in which respect their work is different compared to previous studies. The rest thing I am finding is ok.

Author Response

We thank the reviewer for their time and helpful comments - the feedback has greatly improved this article. Below is our respons to comment. All changes to the text of the article are color-coded: green is what has been added; yellow is what is to be deleted.

The method of free damping torsion oscillations for the first time obtained spectra with a fine complex structure, which characterizes the distributions over the temperature regions of local dissipative processes. The earlier works are characterized by blurred maxima in the regions of the most pronounced dissipative processes, as the experimental points in them were obtained in large temperature intervals (10 - 20 K).

In addition, in the course of the experiment the temperature dependences of the frequency of the freely damped oscillatory process were obtained for the first time and a theoretical analysis of the relationship between the anomalous change in the frequency of the oscillatory process and the shear modulus defect was performed.

Using the obtained experimental temperature dependences of the frequency of free damped torsional oscillations excited in the PE samples under study during their theoretical treatment, the following physical-chemical and physical-mechanical characteristics of the polymer were determined for the first time: the shear modulus defect value for each of the local dissipative processes found on the internal friction spectrum; the internal friction mechanism of these processes; the real temperature decrease of the shear modulus, taking into account the effect of local defect on the value of this modulus.

Thus, comparing experimental and calculated data, it is possible to make a prediction about the behavior of polymer material in a wide range of temperatures: - from conditions of negative temperatures (operation of products in the northern Arctic regions) to high temperatures (above the upper limit of the general industrial temperature range of operation).

Reviewer 3 Report

Comments and Suggestions for Authors

The manuscript "Relaxation Phenomena in Low-Density and High-Density Polyethylene" shows deepen analysis using DSC investigating LDPE and HDPE relaxation phenomena. The manuscript needs revision.

1. The introduction is kind of mixed with materials and methods but should more or less give evidence why such investigation needed and what contribution other researchers have done to such phenomena. Please write more clear what is the goal and what purpose serve such investigation?

2. The application of this investigation should be strength either modulus or tensile strength but none of those measurements been presented here. Please include those hence its the most differences between LDPE and HDPE

3. If knowing those relaxation phenomena how can the synthesis of LDPE or HDPE improved? Are such investigation made for other plastics in the past and how does this results contributes to a better understanding.

Comments on the Quality of English Language

Minor spell checking needed throughout the manuscript

Author Response

(The authors gave the same response as above.)
